# HUMUS-Net: Hybrid Unrolled Multi-Scale Network Architecture for Accelerated MRI Reconstruction

**Zalan Fabian**
Dept. of Electrical and Computer Engineering
University of Southern California
Los Angeles, CA
zfabian@usc.edu

**Berk Tinaz**
Dept. of Electrical and Computer Engineering
University of Southern California
Los Angeles, CA
tinaz@usc.edu

**Mahdi Soltanolkotabi**
Dept. of Electrical and Computer Engineering
University of Southern California
Los Angeles, CA
soltanol@usc.edu

## Abstract

In accelerated MRI reconstruction, the anatomy of a patient is recovered from a set of under-sampled and noisy measurements. Deep learning approaches have been proven to be successful in solving this ill-posed inverse problem and are capable of producing very high quality reconstructions. However, current architectures heavily rely on convolutions, that are content-independent and have difficulties modeling long-range dependencies in images. Recently, Transformers, the workhorse of contemporary natural language processing, have emerged as powerful building blocks for a multitude of vision tasks. These models split input images into non-overlapping patches, embed the patches into lower-dimensional tokens and utilize a self-attention mechanism that does not suffer from the aforementioned weaknesses of convolutional architectures. However, Transformers incur extremely high compute and memory cost when 1) the input image resolution is high and 2) when the image needs to be split into a large number of patches to preserve fine detail information, both of which are typical in low-level vision problems such as MRI reconstruction, having a compounding effect. To tackle these challenges, we propose HUMUS-Net, a hybrid architecture that combines the beneficial implicit bias and efficiency of convolutions with the power of Transformer blocks in an unrolled and multi-scale network. HUMUS-Net extracts high-resolution features via convolutional blocks and refines low-resolution features via a novel Transformer-based multi-scale feature extractor. Features from both levels are then synthesized into a high-resolution output reconstruction. Our network establishes new state of the art on the largest publicly available MRI dataset, the fastMRI dataset. We further demonstrate the performance of HUMUS-Net on two other popular MRI datasets and perform fine-grained ablation studies to validate our design.

## 1 Introduction

Magnetic resonance imaging (MRI) is a medical imaging technique that uses strong magnetic fields to picture the anatomy and physiological processes of the patient. MRI is one of the most popular imaging modalities as it is non-invasive and doesn't expose the patient to harmful ionizing radiation. The MRI scanner obtains measurements of the body in the spatial frequency domain also called

36th Conference on Neural Information Processing Systems (NeurIPS 2022).

*k-space*. The data acquisition process is MR is often time-consuming. Accelerated MRI [Lustig et al., 2008] addresses this challenge by undersampling in the k-space domain, thus reducing the time patients need to spend in the scanner. However, recovering the underlying anatomy from undersampled measurements is an ill-posed problem (less measurements than unknowns) and thus incorporating some form of prior knowledge is crucial in obtaining high quality reconstructions. Classical MRI reconstruction algorithms rely on the assumption that the underlying signal is sparse in some transform domain and attempt to recover a signal that best satisfies this assumption in a technique known as compressed sensing (CS) [Candes et al., 2006, Donoho, 2006]. These classical CS techniques have slow reconstruction speed and typically enforce limited forms of image priors.

With the emergence of deep learning (DL), data-driven reconstruction algorithms have far surpassed CS techniques (see Ongie et al. [2020] for an overview). DL models utilize large training datasets to extract flexible, nuanced priors directly from data resulting in excellent reconstruction quality. In recent years, there has been a flurry of activity aimed at designing DL architectures tailored to the MRI reconstruction problem. The most popular models are convolutional neural networks (CNNs) that typically incorporate the physics of the MRI reconstruction problem, and utilize tools from mainstream deep learning (residual learning, data augmentation, self-supervised learning). Comparing the performance of such models has been difficult mainly due to two reasons. First, there has been a large variation in evaluation datasets spanning different scanners, anatomies, acquisition models and undersampling patterns rendering direct comparison challenging. Second, medical imaging datasets are often proprietary due to privacy concerns, hindering reproducibility.

More recently, the fastMRI dataset [Zbontar et al., 2019], the largest publicly available MRI dataset, has been gaining ground as a standard benchmark to evaluate MRI reconstruction methods. An annual competition, the fastMRI Challenge [Muckley et al., 2021], attracts significant attention from the machine learning community and acts a driver of innovation in MRI reconstruction. However, over the past years the public leaderboard has been dominated by a single architecture, the End-to-End VarNet [Sriram et al., 2020] [1], with most models concentrating very closely around the same performance metrics, hinting at the saturation of current architectural choices.

In this work, we propose HUMUS-Net: a Hybrid, Unrolled, MUlti-Scale network architecture for accelerated MRI reconstruction that combines the advantages of well-established architectures in the field with the power of contemporary Transformer-based models. We utilize the strong implicit bias of convolutions, but also address their weaknesses, such as content-independence and inability to model long-range dependencies, by incorporating a novel multi-scale feature extractor that operates over embedded image patches via self-attention. Moreover, we tackle the challenge of high input resolution typical in MRI by performing the computationally most expensive operations on extracted low-resolution features. HUMUS-Net establishes new state of the art in accelerated MRI reconstruction on the largest available MRI knee dataset. At the time of writing this paper, HUMUS-Net is the only Transformer-based architecture on the highly competitive fastMRI Public Leaderboard. Our results are fully reproducible and the source code is available online [2]

## 2 Background

### 2.1 Inverse Problem Formulation of Accelerated MRI Reconstruction

An MR scanner obtains measurements of the patient anatomy in the frequency domain, also referred to as *k-space*. Data acquisition is performed via various receiver coils positioned around the anatomy being imaged, each with different spatial sensitivity. Given a total number of $N$ receiver coils, the measurements obtained by the $i$th coil can be written as

$$\boldsymbol{k_i} = \mathcal{F}\boldsymbol{S_i}\boldsymbol{x}^* + \boldsymbol{z_i}, \ \ i = 1, .., N,$$

where $\boldsymbol{x}^* \in \mathbb{C}^n$ is the underlying patient anatomy of interest, $\boldsymbol{S_i}$ is a diagonal matrix that represents the sensitivity map of the $i$th coil, $\mathcal{F}$ is a multi-dimensional Fourier-transform, and $z_i$ denotes the measurement noise corrupting the observations obtained from coil $i$. We use $\boldsymbol{k} = (\boldsymbol{k_1}, ..., \boldsymbol{k_N})$ as

---

[1]At the time of writing this paper, the number one model on the leaderboard is AIRS-Net, however the complete architecture/training details as well as the code of this method are not available to the public and we were unable to reproduce their results based on the limited available information.

[2]Code: https://github.com/MathFLDS/HUMUS-Net.

a shorthand for the concatenation of individual coil measurements and $\boldsymbol{x} = (\boldsymbol{x}_1, ..., \boldsymbol{x}_N)$ as the corresponding image domain representation after inverse Fourier transformation.

Since MR data acquisition time is proportional to the portion of k-space being scanned, obtaining fully-sampled data is time-consuming. Therefore, in *accelerated MRI* scan times are reduced by undersampling in k-space domain. The undersampled k-space measurements from coil $i$ take the form

$$\tilde{\boldsymbol{k}_i} = \boldsymbol{M}\boldsymbol{k_i} \ \ i = 1, .., N,$$

where $\boldsymbol{M}$ is a diagonal matrix representing the binary undersampling mask, that has $0$ values for all missing frequency components that have not been sampled during accelerated acquisition.

The forward model that maps the underlying anatomy to coil measurements can be written concisely as $\tilde{\boldsymbol{k}} = \mathcal{A}(\boldsymbol{x}^*)$, where $\mathcal{A}(\cdot)$ is the linear forward mapping and $\tilde{\boldsymbol{k}}$ is the stacked vector of all undersampled coil measurements. Our target is to reconstruct the ground truth object $\boldsymbol{x}^*$ from the noisy, undersampled measurements $\tilde{\boldsymbol{k}}$. Since we have fewer observations than variables to recover, perfect reconstruction in general is not possible. In order to make the problem solvable, prior knowledge on the underlying object is typically incorporated in the form of sparsity in some transform domain. This formulation, known as compressed sensing [Candes et al., 2006, Donoho, 2006], provides a classical framework for accelerated MRI reconstruction [Lustig et al., 2008]. In particular, the above recovery problem can be formulated as a regularized inverse problem

$$\hat{\boldsymbol{x}} = \arg\min_{\boldsymbol{x}} \left\| \mathcal{A}(\boldsymbol{x}) - \tilde{\boldsymbol{k}} \right\|^2 + \mathcal{R}(\boldsymbol{x}), \tag{2.1}$$

where $\mathcal{R}(\cdot)$ is a regularizer that encapsulates prior knowledge on the object, such as sparsity in some wavelet domain.

## 2.2 Deep Learning-based Accelerated MRI Reconstruction

More recently, data-driven deep learning-based algorithms tailored to the accelerated MRI reconstruction problem have surpassed the classical compressed sensing baselines. Convolutional neural networks trained on large datasets have established new state of the art in many medical imaging tasks. The highly popular U-Net [Ronneberger et al., 2015] and other similar encoder-decoder architectures have proven to be successful in a range of medical image reconstruction [Hyun et al., 2018, Han and Ye, 2018] and segmentation [Çiçek et al., 2016, Zhou et al., 2018] problems. In the encoder path, the network learns to extract a set of deep, low-dimensional features from images via a series of convolutional and downsampling operations. These concise feature representations are then gradually upsampled and filtered in the decoder to the original image dimensions. Thus the network learns a hierarchical representation over the input image distribution.

Unrolled networks constitute another line of work that has been inspired by popular optimization algorithms used to solve compressed sensing reconstruction problems. These deep learning models consist of a series of sub-networks, also known as cascades, where each sub-network corresponds to an unrolled iteration of popular algorithms such as gradient descent [Zhang and Ghanem, 2018] or ADMM [Sun et al., 2016]. In the context of MRI reconstruction, one can view network unrolling as solving a sequence of smaller denoising problems, instead of the complete recovery problem in one step. Various convolutional neural networks have been employed in the unrolling framework achieving excellent performance in accelerated MRI reconstruction [Putzky et al., 2019, Hammernik et al., 2018, 2019]. E2E-VarNet [Sriram et al., 2020] is the current state-of-the-art convolutional model on the fastMRI dataset. E2E-VarNet transforms the optimization problem in (2.1) to the k-space domain and unrolls the gradient descent iterations into $T$ cascades, where the $t$th cascade represents the computation

$$\hat{\boldsymbol{k}}^{t+1} = \hat{\boldsymbol{k}}^t - \mu^t \boldsymbol{M}(\hat{\boldsymbol{k}}^t - \tilde{\boldsymbol{k}}) + \mathcal{G}(\hat{\boldsymbol{k}}^t), \tag{2.2}$$

where $\hat{\boldsymbol{k}}^t$ is the estimated reconstruction in the k-space domain at cascade $t$, $\mu^t$ is a learnable step size parameter and $\mathcal{G}(\cdot)$ is a learned mapping representing the gradient of the regularization term in (2.1). The first term is also known as *data consistency* (DC) term as it enforces the consistency of the estimate with the available measurements.

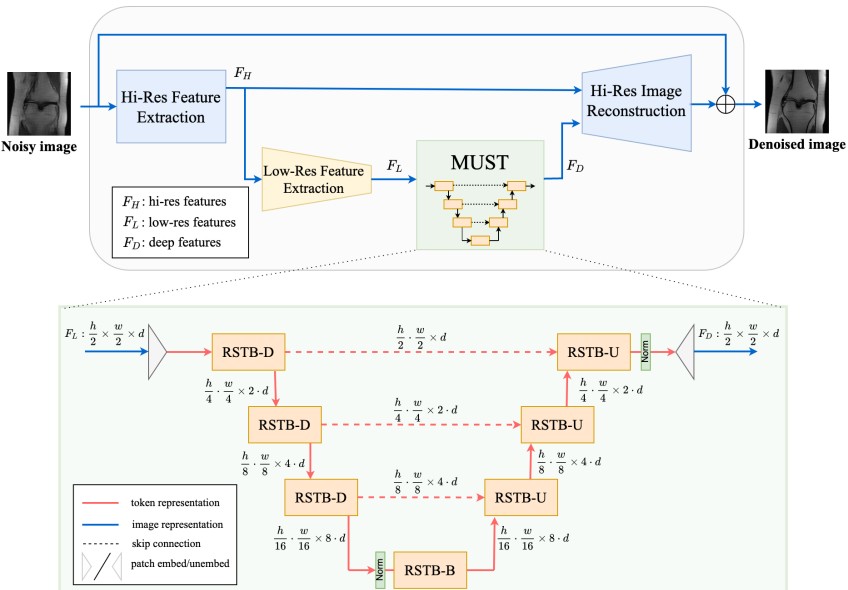

Figure 1: Overview of the HUMUS-Block architecture. First, we extract high-resolution features $\boldsymbol{F_H}$ from the input noisy image through a convolution layer $f_H$. Then, we apply a convolutional feature extractor $f_L$ to obtain low-resolution features and process them using a Transformer-convolutional hybrid multi-scale feature extractor. The shallow, high-resolution and deep, low-resolution features are then synthesized into the final high-resolution denoised image.

## 3 Related Work

**Transformers in Vision –** Vision Transformer (ViT) [Dosovitskiy et al., 2020], a fully non-convolutional vision architecture, has demonstrated state-of-the-art performance on image classification problems when pre-trained on large-scale image datasets. The key idea of ViT is to split the input image into non-overlapping patches, embed each patch via a learned linear mapping and process the resulting tokens via stacked self-attention and multi-layer perceptron (MLP) blocks. For more details we refer the reader to the supplementary and [Dosovitskiy et al., 2020]. The benefit of Transformers over convolutional architectures in vision lies in their ability to capture long-range dependencies in images via the self-attention mechanism.

Since the introduction of ViT, similar attention-based architectures have been proposed for many other vision tasks such as object detection [Carion et al., 2020], image segmentation [Wang et al., 2021b] and restoration [Cao et al., 2021b, Chen et al., 2021, Liang et al., 2021, Zamir et al., 2021, Wang et al., 2021c]. A key challenge for Transformers in low-level vision problems is the quadratic compute complexity of the global self-attention with respect to the input dimension. In some works, this issue has been mitigated by splitting the input image into fixed size patches and processing the patches independently [Chen et al., 2021]. Others focus on designing hierarchical Transformer architectures [Heo et al., 2021, Wang et al., 2021b] similar to popular ResNets [He et al., 2015]. Authors in Zamir et al. [2021] propose applying self-attention channel-wise rather than across the spatial dimension thus reducing the compute overhead to linear complexity. Another successful architecture, the Swin Transformer [Liu et al., 2021], tackles the quadratic scaling issue by computing self-attention on smaller local windows. To encourage cross-window interaction, windows in subsequent layers are shifted relative to each other.

**Transformers in Medical Imaging –** Transformer architectures have been proposed recently to tackle various medical imaging problems. Authors in Cao et al. [2021a] design a U-Net-like architecture for medical image segmentation where the traditional convolutional layers are replaced by Swin Transformer blocks. They report strong results on multi-organ and cardiac image segmentation. In Zhou et al. [2021] a hybrid convolutional and Transformer-based U-Net architecture is proposed tailored to volumetric medical image segmentation with excellent results on benchmark datasets. Similar encoder-decoder architectures for various medical segmentation tasks have been investigated in other works [Huang et al., 2021, Wu et al., 2022]. However, these networks are tailored for image

segmentation, a task less sensitive to fine details in the input, and thus larger patch sizes are often used (for instance 4 in Cao et al. [2021a] ). This allows the network to process larger input images, as the number of token embedding is greatly reduced, but as we demonstrate in Section 5.2 embedding individual pixels as $1 \times 1$ patches is crucial for MRI reconstruction. Thus, compute and memory barriers stemming from large input resolutions are more severe in the MRI reconstruction task and therefore novel approaches are needed.

Promising results have been reported employing Transformers in medical image denoising problems, such as low-dose CT denoising [Wang et al., 2021a, Luthra et al., 2021] and low-count PET/MRI denoising [Zhang et al., 2021]. However, these studies fail to address the challenge of poor scaling to large input resolutions, and only work on small images via either downsampling the original dataset [Luthra et al., 2021] or by slicing the large input images into smaller patches [Wang et al., 2021a]. In contrast, our proposed architecture works directly on the large resolution images that often arise in MRI reconstruction. Even though some work exists on Transformer-based architectures for supervised accelerated MRI reconstruction [Huang et al., 2022, Lin and Heckel, 2022, Feng et al., 2021], and for unsupervised pre-trained reconstruction [Korkmaz et al., 2022], to the best of our knowledge ours is the first work to demonstrate state-of-the-art results on large-scale MRI datasets such as the fastMRI dataset.

# 4    Method

HUMUS-Net combines the efficiency and beneficial implicit bias of convolutional networks with the powerful general representations of Transformers and their capability to capture long-range pixel dependencies. The resulting hybrid network processes information both in image representation (via convolutions) and in patch-embedded token representation (via Transformer blocks). Our proposed architecture consists of a sequence of sub-networks, also called cascades. Each cascade represents an unrolled iteration of an underlying optimization algorithm in k-space (see (2.2)), with an image-domain denoiser, the HUMUS-Block. First, we describe the architecture of the HUMUS-Block, the core component in the sub-network. Then, we specify the high-level, k-space unrolling architecture of HUMUS-Net in Section 4.3.

## 4.1    HUMUS-Block Architecture

The HUMUS-Block acts as an image-space denoiser that receives an intermediate reconstruction from the previous cascade and performs a single step of denoising to produce an improved reconstruction for the next cascade. It extracts high-resolution, shallow features and low-resolution, deep features through a novel multi-scale transformer-based block, and synthesizes high-resolution features from those. The high-level overview of the HUMUS-Block is depicted in Fig. 1.

**High-resolution Feature Extraction–** The input to our network is a noisy complex-valued image $\boldsymbol{x}_{in} \in \mathbb{R}^{h \times w \times c_{in}}$ derived from undersampled k-space data, where the real and imaginary parts of the image are concatenated along the channel dimension. First, we extract high-resolution features $\boldsymbol{F_H} \in \mathbb{R}^{h \times w \times d_H}$ from the input noisy image through a convolution layer $f_H$ written as $\boldsymbol{F_H} = f_H(\boldsymbol{x}_{in})$. This initial $3 \times 3$ convolution layer provides early visual processing at a relatively low cost and maps the input to a higher, $d_H$ dimensional feature space. Important to note that the resolution of the extracted features is the same as the spatial resolution of the input image.

**Low-resolution Feature Extraction–** In case of MRI reconstruction, input resolution is typically significantly higher than commonly used image datasets ($32 \times 32$ - $256 \times 256$), posing a significant challenge to contemporary Transformer-based models. Thus we apply a convolutional feature extractor $f_L$ to obtain low-resolution features $\boldsymbol{F_L} = f_L(\boldsymbol{F_H})$, with $\boldsymbol{F_L} \in \mathbb{R}^{h_L \times w_L \times d_L}$ where $f_L$ consists of a sequence of convolutional blocks and spatial downsampling operations. The specific architecture is depicted in Figure 4. The purpose of this module is to perform deeper visual processing and to provide manageable input size to the subsequent computation and memory heavy hybrid processing module. In this work, we choose $h_L = \frac{h}{2}$ and $w_L = \frac{w}{2}$ which strikes a balance between preserving spatial information and resource demands. Furthermore, in order to compensate for the reduced resolution we increase the feature dimension to $d_L = 2 \cdot d_H := d$.

**Deep Feature Extraction–** The most important part of our model is MUST, a **MU**lti-scale residual **S**win **T**ransformer network. MUST is a multi-scale hybrid feature extractor that takes the low-

resolution image representations $\boldsymbol{F_L}$ and performs hierarchical Transformer-convolutional hybrid processing in an encoder-decoder fashion, producing deep features $\boldsymbol{F_D} = f_D(\boldsymbol{F_L})$, where the specific architecture behind $f_D$ is detailed in Subsection 4.2.

**High-resolution Image Reconstruction–** Finally, we combine information from shallow, high-resolution features $\boldsymbol{F_H}$ and deep, low-resolution features $\boldsymbol{F_D}$ to reconstruct the high-resolution residual image via a convolutional reconstruction module $f_R$. The residual learning paradigm allows us to learn the difference between noisy and clean images and helps information flow within the network [He et al., 2015]. Thus the final denoised image $\boldsymbol{x}_{out} \in \mathbb{R}^{h \times w \times c_{in}}$ is obtained as $\boldsymbol{x}_{out} = \boldsymbol{x}_{in} + f_R(\boldsymbol{F_H}, \boldsymbol{F_D})$. The specific architecture of the reconstruction network is depicted in Figure 4.

## 4.2   Multi-scale Hybrid Feature Extraction via MUST

The key component to our architecture is MUST, a multi-scale hybrid encoder-decoder architecture that performs deep feature extraction in both image and token representation (Figure 1, bottom). First, individual pixels of the input representation of shape $\frac{h}{2} \times \frac{w}{2} \times d$ are flattened and passed through a learned linear mapping to yield $\frac{h}{2} \cdot \frac{w}{2}$ tokens of dimension $d$. Tokens corresponding to different image patches are subsequently merged in the encoder path, resulting in a concise latent representation. This highly descriptive representation is passed through a bottleneck block and progressively expanded by combining tokens from the encoder path via skip connections. The final output is rearranged to match the exact shape of the input low-resolution features $\boldsymbol{F_L}$, yielding a deep feature representation $\boldsymbol{F_D}$.

Our design is inspired by the success of Residual Swin Transformer Blocks (RSTB) in image denoising and super-resolution [Liang et al., 2021]. RSTB features a stack of Swin Transformer layers (STL) that operate on tokens via a windowed self-attention mechanism [Liu et al., 2021], followed by convolution in image representation. However, RSTB blocks operate on a single scale, therefore they cannot be readily applied in a hierarchical encoder-decoder architecture. Therefore, we design three variations of RSTB to facilitate multi-scale processing as depicted in Figure 2.

**RSTB-B** is the bottleneck block responsible for processing the encoded latent representation while maintaining feature dimensions. Thus, we keep the default RSTB architecture for our bottleneck block, which already operates on a single scale.

**RSTB-D** has a similar function to convolutional downsampling blocks in U-Nets, but it operates on embedded tokens. Given an input with size $h_i \cdot w_i \times d$, we pass it through an RSTB-B block and apply *PatchMerge* operation. *PatchMerge* linearly combines tokens corresponding to $2 \times 2$ non-overlapping image patches, while simultaneously increasing the embedding dimension (see Figure 3, top and the supplementary for more details) resulting in an output of size $\frac{h_i}{2} \cdot \frac{w_i}{2} \times 2 \cdot d$. Furthermore, RSTB-D outputs the higher dimensional representation before patch merging to be subsequently used in the decoder path via skip connection.

**RSTB-U** used in the decoder path is analogous to convolutional upsampling blocks. An input with size $h_i \cdot w_i \times d$ is first expanded into a larger number of lower dimensional tokens through a linear mapping via *PatchExpand* (see Figure 3, bottom and the supplementary for more details). *PatchExpand* reverses the effect of *PatchMerge* on feature size, thus resulting in $2h_i \cdot 2w_i$ tokens of dimension $\frac{d}{2}$. Next, we mix information from the obtained expanded tokens with skip embeddings from higher scales via *TokenMix*. This operation linearly combines tokens from both paths and normalizes the resulting vectors. Finally, the mixed tokens are processed by an RSTB-B block.

## 4.3   Iterative Unrolling

Architectures derived from unrolling the iterations of various optimization algorithms have proven to be successful in tackling various inverse problems including MRI reconstruction. These architecture can be interpreted as a cascade of simpler denoisers, each of which progressively refines the estimate from the preceding unrolled iteration (see more details in the supplementary).

Following Sriram et al. [2020], we unroll the gradient descent iterations of the inverse problem in (2.1) in k-space domain, yielding the iterative update scheme in (2.2). We apply regularization in image domain via our proposed HUMUS-Block, that is we have $\mathcal{G}(\boldsymbol{k}) = \mathcal{F}\left(\mathcal{E}\left(\mathbf{D}\left(\mathcal{R}\left(\mathcal{F}^{-1}(\boldsymbol{k})\right)\right)\right)\right)$, where $\mathbf{D}$ denotes the HUMUS-Block, $\mathcal{R}(\boldsymbol{x_1}, ..., \boldsymbol{x_N}) = \sum_{i=1}^{N} \boldsymbol{S}_i^* x_i$ is the *reduce* operator that combines

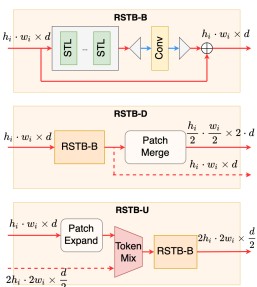 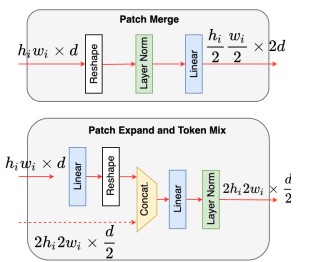 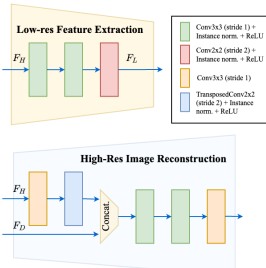

Figure 2: Depiction of different RSTB modules used in the HUMUS-Block.

Figure 3: Patch merge and expand operations used in our multi-scale feature extractor.

Figure 4: Architecture of convolutional blocks for feature extraction and reconstruction.

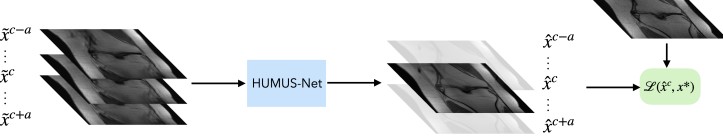

Figure 5: Adjacent slice reconstruction (depicted in image domain for visual clarity): HUMUS-Net takes a volume of adjacent slices $(\tilde{\boldsymbol{x}}^{c-a}, ..., \tilde{\boldsymbol{x}}^c, ..., \tilde{\boldsymbol{x}}^{c+a})$ and jointly reconstructs a volume $(\hat{\boldsymbol{x}}^{c-a}, ..., \hat{\boldsymbol{x}}^c, ..., \hat{\boldsymbol{x}}^{c+a})$. The reconstruction loss $\mathcal{L}$ is calculated only on the center slice $\hat{\boldsymbol{x}}^c$.

coil images via the corresponding sensitivity maps and $\mathcal{E}(\boldsymbol{x}) = (\boldsymbol{S}_1\boldsymbol{x}, ..., \boldsymbol{S}_N\boldsymbol{x})$ is the *expand* operator that maps the combined image back to individual coil images. The sensitivity maps can be estimated a priori using methods such as ESPIRiT [Uecker et al., 2014] or learned in an end-to-end fashion via a Sensitivity Map Estimator (SME) network proposed in Sriram et al. [2020]. In this work we aspire to design an end-to-end approach and thus we use the latter method and estimate the sensitivity maps from the low-frequency (ACS) region of the undersampled input measurements during training using a standard U-Net network.

### 4.4 Adjacent Slice Reconstruction (ASR)

We observe improvements in reconstruction quality when instead of processing the undersampled data slice-by-slice, we jointly reconstruct a set of adjacent slices via HUMUS-Net (Figure 5). That is, if we have a volume of undersampled data $\tilde{\boldsymbol{k}}^{vol} = (\tilde{\boldsymbol{k}}^1, ..., \tilde{\boldsymbol{k}}^K)$ with $K$ slices, when reconstructing slice $c$, we instead reconstruct the volume $(\tilde{\boldsymbol{k}}^{c-a}, ..., \tilde{\boldsymbol{k}}^{c-1}, \tilde{\boldsymbol{k}}^c, \tilde{\boldsymbol{k}}^{c+1}, ..., \tilde{\boldsymbol{k}}^{c+a})$ by concatenating them along the coil channel dimension, where $a$ denotes the number of adjacent slices added on each side. However, we only calculate and backpropagate the loss on the center slice $c$ of the reconstructed volume. The benefit of ASR is that the network can remove artifacts corrupting individual slices as it sees a larger context of the slice by observing its neighbors. Even though ASR increases compute cost, it is important to note that it does not impact the number of token embeddings (spatial resolution is unchanged) and thus can be combined favorably with Transformer-based methods.

## 5 Experiments

In this section we provide experimental results on our proposed architecture, HUMUS-Net. First, we demonstrate the reconstruction performance of our model on various datasets, including the large-scale fastMRI dataset. Then, we justify our design choices through a set of ablation studies.

### 5.1 Benchmark Experiments

We investigate the performance of HUMUS-Net on three different datasets. We use the structural similarity index measure (SSIM) [Wang et al., 2003] as a basis of our evaluation, which is the most common evaluation metric in medical image reconstruction. In all of our experiments, we follow the setup of the fastMRI multi-coil knee track with $8\times$ acceleration in order to provide comparison with state-of-the-art networks of the Public Leaderboard. That is, we perform retrospective undersampling of the fully-sampled k-space data by randomly subsampling $12.5\%$ of whole k-space lines in the

| fastMRI knee multi-coil ×8 test data | | | | |
|---|---|---|---|---|
| **Method** | **# of params** (approx.) | **SSIM(↑)** | **PSNR(↑)** | **NMSE(↓)** |
| E2E-VarNet [Sriram et al., 2020] | 30M | 0.8900 | 36.9 | 0.0089 |
| E2E-VarNet† [Sriram et al., 2020] | | 0.8920 | 37.1 | 0.0085 |
| XPDNet [Ramzi et al., 2020] | 155M | 0.8893 | 37.2 | 0.0083 |
| Σ-Net [Hammernik et al., 2019] | 676M | 0.8877 | 36.7 | 0.0091 |
| i-RIM [Putzky et al., 2019] | 300M | 0.8875 | 36.7 | 0.0091 |
| U-Net [Zbontar et al., 2019] | 214M | 0.8640 | 34.7 | 0.0132 |
| HUMUS-Net (ours) | 109M | 0.8936 | 37.0 | 0.0086 |
| HUMUS-Net (ours)† | | 0.8945 | 37.3 | 0.0081 |
| HUMUS-Net-L (ours) | 228M | 0.8944 | 37.3 | 0.0083 |
| HUMUS-Net-L (ours)† | | **0.8951** | **37.4** | **0.0080** |

Table 1: Performance of state-of-the-art accelerated MRI reconstruction techniques on the fastMRI knee test dataset. Most models are trained only on the fastMRI training dataset, if available we show results of models trained on the fastMRI combined training and validation dataset denoted by (†).

| Validation SSIM | | | |
|---|---|---|---|
| **Method** | **fastMRI knee** | **Stanford 2D** | **Stanford 3D** |
| E2E-VarNet [Sriram et al., 2020] | 0.8908 | $0.8928 \pm 0.0168$ | $0.9432 \pm 0.0063$ |
| HUMUS-Net (ours) | **0.8934** | **$0.8954 \pm 0.0136$** | **$0.9453 \pm 0.0065$** |

Table 2: Validation SSIM of HUMUS-Net on various datasets. For datasets with multiple train-validation split runs we show the mean and standard error of the runs.

| Ablation studies | | | | | | |
|---|---|---|---|---|---|---|
| **Method** | **Unrolled?** | **Multi-scale?** | **Low-res features?** | **Patch size** | **Embed. dim.** | **SSIM** |
| Un-SS | ✓ | ✗ | ✗ | 1 | 12 | $0.9319 \pm 0.0080$ |
| Un-MS | ✓ | ✓ | ✗ | 1 | 12 | $0.9357 \pm 0.0038$ |
| Un-MS-Patch2 | ✓ | ✓ | ✗ | 2 | 36 | $0.9171 \pm 0.0075$ |
| HUMUS-Net | ✓ | ✓ | ✓ | 1 | 36 | **$0.9449 \pm 0.0064$** |
| SwinIR | | | | | | $0.9336 \pm 0.0069$ |
| E2E-VarNet | | | | | | $0.9432 \pm 0.0063$ |

Table 3: Results of ablation studies on HUMUS-Net, evaluated on the Stanford 3D MRI dataset.

phase encoding direction, keeping $4\%$ of lowest frequency adjacent k-space lines. Experiments on other acceleration ratios can be found in the supplementary. During training, we generate random masks following the above method, whereas for the validation dataset we keep the masks fixed for each k-space volume. For HUMUS-Net, we center crop and pad inputs to $384 \times 384$. We compare the reconstruction quality of our proposed model with the current best performing network, E2E-VarNet. For details on HUMUS-Net hyperparameters and training, we refer the reader to the supplementary. For E2E-VarNet, we use the hyperparameters specified in Sriram et al. [2020].

**fastMRI –** The fastMRI dataset [Zbontar et al., 2019] is the largest publicly available MRI dataset with competitive baseline models and a public leaderboard, and thus provides an opportunity to directly compare different algorithms. Specifically, we run experiments on the multi-coil knee dataset, consisting of close to $35k$ slices in $973$ volumes. We use the default HUMUS-Net model defined above with 3 adjacent slices as input. Furthermore, we design a large variant of our model, *HUMUS-Net-L*, which has increased embedding dimension compared to the default model (see details in the supplementary). We train models both only on the training split, and also on the training and validation splits combined (additional $\approx 20\%$ data) for the leaderboard. Table 1 demonstrates our results compared to the best published models from the fastMRI Leaderboard[3] evaluated on the test dataset. Our model establishes new state of the art in terms of SSIM on this dataset by a large margin, and achieves comparable or better performance than other methods in terms of PSNR and NMSE. Moreover, as seen in the second column of Table 2, we evaluated our model on the fastMRI validation dataset as well and compared our results to E2E-VarNet, the best performing model from the leaderboard. We observe similar improvements in terms of the reconstruction SSIM metric to the test dataset. Visual inspection of reconstructions shows that HUMUS-Net recovers very fine details in images that may be missed by other state-of-the-art reconstruction algorithms (see attached

---

[3]https://fastmri.org/leaderboards

figures in the supplementary). Comparison based on further image quality metrics can be found in the supplementary. We point out that even though our model has more parameters than E2E-VarNet, our proposed image-domain denoiser is more efficient than the U-Net deployed in E2E-VarNet, even when their number of parameters are matched as discussed in Section 5.3. Overall, larger model size does not necessarily correlate with better performance, as seen in other competitive models in Table 1. Finally, we note that the additional training data (in the form of the validation split) provides a consistent small boost to model performance. We refer the reader to Klug and Heckel [2022] for an overview of scaling properties of reconstruction models.

**Stanford 2D –** Next, we run experiments on the Stanford2D FSE [Cheng] dataset, a publicly available MRI dataset consisting of scans from various anatomies (pelvis, lower extremity and more) in 89 fully-sampled volumes. We randomly sample $80\%$ of volumes as train data and use the rest for validation. We randomly generate 3 different train-validation splits this way to reduce variations in the presented metrics. As slices in this dataset have widely varying shapes across volumes, we center crop the target images to keep spatial resolution within $384 \times 384$. We use the default HUMUS-Net defined above with single slices as input. Our results comparing the best performing MRI reconstruction model with HUMUS-Net is shown in the third column of Table 2. We present the mean SSIM of all runs along with the standard error. We achieve improvements of similar magnitude as on the fastMRI dataset. These results demonstrate the effectiveness of HUMUS-Net on a more diverse dataset featuring multiple anatomies.

**Stanford 3D –** Finally, we evaluate our model on the Stanford Fullysampled 3D FSE Knees dataset [Sawyer et al., 2013], a public MRI dataset including 20 volumes of knee MRI scans. We generate train-validation splits using the method described for Stanford 2D and perform 3 runs. We use the default HUMUS-Net network with single slices as input. The last column of Table 2 compares our results to E2E-VarNet, showing improvements of similar scales as on other datasets we have investigated in this work. This experiment demonstrates that HUMUS-Net performs well not only on large-scale MRI datasets, but also on smaller problems.

## 5.2 Ablation Studies

In this section, we motivate our design choices through a set of ablation studies. We start from SwinIR, a general image reconstruction network, and highlight its weaknesses for MRI. Then, we demonstrate step-by-step how we addressed these shortcomings and arrived at the HUMUS-Net architecture. We train the models on the Stanford 3D dataset. More details can be found in the supplementary. The results of our ablation studies are summarized in Table 3.

First, we investigate SwinIR, a state-of-the-art image denoiser and super-resolution model that features a hybrid Transformer-convolutional architecture. In order to handle the $10\times$ larger input sizes ($320 \times 320$) in our MRI dataset compared to the input images this network has been designed for ($128 \times 128$), we reduce the embedding dimension of SwinIR to fit into GPU memory (16 GB). We find that compared to models designed for MRI reconstruction, such as E2E-VarNet (last row in Table 3) SwinIR performs poorly. This is not only due to the reduced network size, but also due to the fact that SwinIR is not specifically designed to take the MRI forward model into consideration.

Next, we unroll SwinIR and add a sensitivity map estimator. We refer to this model as *Un-SS*. Due to unrolling, we have to further reduce the embedding dimension of the denoiser and also decrease the depth of the network in order to fit into GPU memory. *Un-SS*, due to its small size, performs slightly worse than vanilla SwinIR and significantly lags behind the E2E-VarNet architectures. We note that SwinIR operates over a single, full-resolution scale, whereas state-of-the-art MRI reconstruction models typically incorporate multi-scale processing in the form of U-Net-like architectures.

Thus, we replace SwinIR by MUST, our proposed multi-scale hybrid processing unit, but keep the embedding dimension in the largest-resolution scale fixed. The obtained network, which we call *Un-MS*, has overall lower computational cost when compared with *Un-SS*, however as Table 3 shows MRI reconstruction performance has significantly improved compared to both *Un-SS* and vanilla SwinIR, which highlights the efficiency of our proposed multi-scale feature extractor. Reconstruction performance is limited by the low dimension of patch embeddings, which we are unable to increase further due to our compute and memory constraints originating in the high-resolution inputs.

The most straightforward approach to tackle the challenge of high input resolution is to increase the patch size. To test this idea, we take *Un-MS* and embed $2 \times 2$ patches of the inputs, thus reducing the

number of tokens processed by the network by a factor of 4. We refer to this model as *Un-MS-Patch2*. This reduction in compute and memory load allows us to increase network capacity by increasing the embedding dimension 3-folds (to fill GPU memory again). However, *Un-MS-Patch2* performs *much* worse than previous models using patch size of 1. For classification problems, where the general image context is more important than small details, patches of $16 \times 16$ or $8 \times 8$ are considered typical [Dosovitskiy et al., 2020]. Even for more dense prediction tasks such as medical image segmentation, patch size of $4 \times 4$ has been used successfully [Cao et al., 2021a]. However, our experiments suggest that in low-level vision tasks such as MRI reconstruction using patches larger than $1 \times 1$ may be detrimental due to loss of crucial high-frequency detail information.

Our approach to address the heavy computational load of Transformers for large input resolutions where increasing the patch size is not an option is to process lower resolution features extracted via convolutions. That is we replace MUST in *Un-MS* by a HUMUS-Block, resulting in our proposed HUMUS-Net architecture. We train a smaller version of HUMUS-Net with the same embedding dimension as *Un-MS*. As seen in Table 3, our model achieves the best performance across all other proposed solutions, even surpassing E2E-VarNet. This series of incremental studies highlights the importance of each architectural design choice leading to our proposed HUMUS-Net architecture.

## 5.3 Direct comparison of image-domain denoisers

In order to further demonstrate the advantage of HUMUS-Net over E2E-VarNet, we provide direct comparison between the image-domain denoisers used in the above methods. E2E-VarNet unrolls a fully convolutional U-Net, whereas we deploy our hybrid HUMUS-Block architecture as a denoiser (Fig. 1). We scale down the HUMUS-Block used in HUMUS-Net to match the size of the U-Net in E2E-Varnet in terms of number of model parameters. To this end, we reduce the embedding dimension from 66 to 30. We train both networks on magnitude images from the Stanford3D dataset. The results are summarized in Table 4. We observe that given a fixed parameter budget, our proposed denoiser outperforms the widely used convolutional U-Net architecture in MRI reconstruction, further demonstarting the efficiency of HUMUS-Net. This experiment suggests that our HUMUS-Block could serve as an excellent denoiser, replacing convolutional U-Nets, in a broad range of image restoration applications outside of MRI, which we leave for future work.

| Model | # of parameters | SSIM($\uparrow$) | PSNR($\uparrow$) | NMSE($\downarrow$) |
|---|---|---|---|---|
| U-Net | 2.5M | $0.9348 \pm 0.0072$ | $39.0 \pm 0.6$ | $0.0257 \pm 0.0007$ |
| HUMUS-Block | 2.4M | $\mathbf{0.9378 \pm 0.0065}$ | $\mathbf{39.2 \pm 0.5}$ | $\mathbf{0.0246 \pm 0.0004}$ |

Table 4: Direct comparison of denoisers on the Stanford 3D dataset. Mean and standard error of 3 random training-validation splits is shown.

## 6 Conclusion

In this paper, we introduce HUMUS-Net, an unrolled, Transformer-convolutional hybrid network for accelerated MRI reconstruction. HUMUS-Net achieves state-of-the-art performance on the fastMRI dataset and greatly outperforms all previous published and reproducible methods. We demonstrate the performance of our proposed method on two other MRI datasets and perform fine-grained ablation studies to motivate our design choices and emphasize the compute and memory challenges of Transformer-based architectures on low-level and dense computer vision tasks such as MRI reconstruction. A limitation of our current architecture is that it requires fixed-size inputs, which we intend to address with a more flexible design in the future. This work opens the door for the adoption of a multitude of promising techniques introduced recently in the literature for Transformers, which we leave for future work.

## Acknowledgments

M. Soltanolkotabi is supported by the Packard Fellowship in Science and Engineering, a Sloan Research Fellowship in Mathematics, an NSF-CAREER under award #1846369, DARPA Learning with Less Labels (LwLL) and FastNICS programs, and NSF-CIF awards #1813877 and #2008443. This research is also in part supported by AWS credits through an Amazon Faculty research award.

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
