# Supplementary Material for HUMUS-Net: Hybrid Unrolled Multi-Scale Network Architecture for Accelerated MRI Reconstruction

## A    HUMUS-Net baseline details

Our default model has 3 RSTB-D downsampling blocks, 2 RSTB-B bottleneck blocks and 3 RSTB-U upsampling blocks with $3 - 6 - 12$ attention heads in the $D/U$ blocks and 24 attention heads in the bottleneck block. For Swin Transformers layers, the window size is 8 for all methods and MLP ratio ($hidden\_dim/input\_dim$) of 2 is used. Each RSTB block consists of 2 STLs with embedding dimension of 66. For HUMUS-Net-L, we increase the embedding dimension to 96. We use 8 cascades of unrolling with a U-Net as sensitivity map estimator (same as in E2E-VarNet) with 16 channels.

We center crop and reflection pad the input images to $384 \times 384$ resolution for HUMUS-Net and use the complete images for VarNet. In all experiments, we minimize the SSIM loss between the target image $x^*$ and the reconstruction $\hat{x}$ defined as

$$\mathcal{L}_{SSIM}(x^*, \hat{x}) = 1 - SSIM(x^*, \hat{x}).$$

**fastMRI experiments–** We train HUMUS-Net using Adam for 50 epochs with a learning rate of 0.0001, dropped by a factor of 10 at epoch 40 and apply adjacent slice reconstruction with 3 slices. We run the experiments on $8\times$ Titan RTX 5000 24GB GPUs with a per-GPU batch size of 1. We train HUMUS-Net-L using a learning rate of 0.00007 (using inverse square root scaling with embedding dimension) for 45 epochs, dropped by a factor of 10 for further 5 epochs. We trained HUMUS-Net-L on $4\times$ A100 40GB GPUs on Amazon SageMaker with per-GPU batch size of 1.

**Stanford 3D experiments–** We train for 25 epochs with a learning rate of 0.0001, dropped by a factor of 10 at epoch 20 and reconstruct single slices. We run the experiments on $4\times$ Quadro RTX 5000 16GB GPUs with a per-GPU batch size of 1.

**Stanford 2D experiments–** We train for 50 epochs with a learning rate of 0.0001, dropped by a factor of 10 at epoch 40 and reconstruct single slices. Moreover, we crop reconstruction targets to fit into a $384 \times 384$ box. We run the experiments on $8\times$ Quadro RTX 5000 16GB GPUs with a per-GPU batch size of 1.

## B    Ablation study experimental details

Here we discuss the experimental setting and hyperparameters used in our ablation study in Section 5.2. In all experiments, we train the models on the Stanford 3D dataset for 3 different train-validation splits and report the mean and standard error of the results. We use Adam optimizer with learning rate 0.0001 and train for 25 epochs, decaying the learning rate by a factor of 10 at 20 epochs. For all unrolled methods, we unroll 12 iterations in order to provide direct comparison with the best performing E2E-VarNet, which uses the same number of cascades. We reconstruct single slices and do not use the method of adjacent slice reconstruction discussed in Section 4.4. In models with sensitivity map estimator,we used the default U-Net with 8 channels (same as default E2E-VarNet). For Swin Transformers layers, the window size is 8 for all methods and MLP ratio ($hidden\_dim/input\_dim$) of 2 is used. Further hyperparameters are summarized in Table 1.

| Method | Embedding dim. | # of STLs in RSTB | # attention heads | Patch size |
|:---:|:---:|:---:|:---:|:---:|
| Un-SS | 12 | $2 - 2 - 2$ | $6 - 6 - 6$ | 1 |
| Un-MS | 12 | $D/U: 2 - 2 - 2, \ B: 2$ | $D/U: 3 - 6 - 12, \ B: 24$ | 1 |
| Un-MS-Patch2 | 36 | $D/U: 2 - 2 - 2, \ B: 2$ | $D/U: 3 - 6 - 12, \ B: 24$ | 2 |
| HUMUS-Net | 36 | $D/U: 2 - 2 - 2, \ B: 2$ | $D/U: 3 - 6 - 12, \ B: 24$ | 1 |
| SwinIR | 66 | $6 - 6 - 6 - 6$ | $6 - 6 - 6 - 6$ | 1 |

Table 1: Ablation study experimental details. We show the number of STL layers per RSTB blocks and number of attention heads for multi-scale networks in downsampling (D) , bottleneck (B) and upsampling (U) paths separately.

We run the experiments on $4\times$ Quadro RTX 5000 16GB GPUs with a per-GPU batch size of 1.

# C   Results on additional accelerations

We observe that HUMUS-Net achieves state-of-the-art performance across a wide range of acceleration ratios. We perform experiments on the Stanford 3D dataset using a small HUMUS-Net model with only 6 cascades, embedding dimension of 66, adjacent slice reconstruction with 3 slices and for the sake of simplicity we removed the residual path from the HUMUS-Block. We call this model *HUMUS-Net-S*. We set the learning rate to 0.0002 and train the model with Adam optimizer. We generate 3 random training-validation splits on the Stanford 3D dataset (same splits as in other Stanford 3D experiments in this paper), and show the mean and standard error of the results in Table 2. In our experiments, HUMUS-Net achieved higher quality reconstructions in every metric over E2E-VarNet.

| Acceleration | Model | SSIM($\uparrow$) | PSNR($\uparrow$) | NMSE($\downarrow$) |
|:---:|:---:|:---:|:---:|:---:|
| 4x | E2E-VarNet | $0.9623 \pm 0.0038$ | $42.9 \pm 0.5$ | $0.0103 \pm 0.0001$ |
| | HUMUS-Net-S | $\mathbf{0.9640 \pm 0.0040}$ | $\mathbf{43.3 \pm 0.5}$ | $\mathbf{0.0096 \pm 0.0002}$ |
| 8x | E2E-VarNet | $0.9432 \pm 0.0063$ | $40.0 \pm 0.6$ | $0.0203 \pm 0.0006$ |
| | HUMUS-Net-S | $\mathbf{0.9459 \pm 0.0065}$ | $\mathbf{40.4 \pm 0.7}$ | $\mathbf{0.0184 \pm 0.0008}$ |
| 12x | E2E-VarNet | $0.9259 \pm 0.0084$ | $37.7 \pm 0.7$ | $0.0347 \pm 0.0018$ |
| | HUMUS-Net-S | $\mathbf{0.9283 \pm 0.0052}$ | $\mathbf{38.0 \pm 0.5}$ | $\mathbf{0.0298 \pm 0.0019}$ |

Table 2: Experiments on various acceleration factors on the Stanford 3D dataset. Mean and standard error of 3 random training-validation splits is shown.

# D   Effect of the number of unrolled iterations

The number of cascades in unrolled networks has a fundamental impact on their performance. Deeper networks typically perform better, but also incur heavy computational and memory cost. In this experiment we investigate the scaling of HUMUS-Net with respect to the number of iterative unrollings.

We perform an ablation study on the Stanford 3D dataset with $\times 8$ acceleration for a fixed training-validation split, where 20% of the training set has been set aside for validation. We reconstruct single slices, without applying adjacent slice reconstruction. We plot the highest SSIM throughout training on the validation dataset for each network on Figure 1.

We observe improvements in reconstruction performance with increasing number of cascades, and this improvement has not saturated yet in the range of model sizes we have investigated. In our main experiments, we select 8 cascades due to memory and compute limitations for deeper models. However, our experiment suggests that HUMUS-Net can potentially obtain even better reconstruction results given enough computational resources.

# E   Effect of Adjacent Slice Reconstruction

In this section, we demonstrate that even though adjacent slice reconstruction is in general helpful, it is not the main reason why HUMUS-Net performs better than E2E-VarNet.

To this end, we add adjacent slice reconstruction to E2E-VarNet and investigate its effect on model performance. We run experiments on the Stanford 3D dataset for 3 different random train-validation splits. We use adjacent slice reconstruction with 3 slices. We have found that the default learning rate for E2E-VarNet is not optimal with the increased input size, therefore we tune the learning rate using grid search and set it to 0.0005. We compare the results with HUMUS-Net, where we match the number of cascades in E2E-VarNet, and use the default embedding dimension of 66. We do not use adjacent slice reconstruction when training HUMUS-Net in order to ablate its effect.

The results are summarized in Table 3. We observe that ASR boosts the reconstruction quality of E2E-VarNet. However, ASR alone cannot close the gap between the two models, as HUMUS-Net without ASR still outperforms the best E2E-VarNet model with ASR.

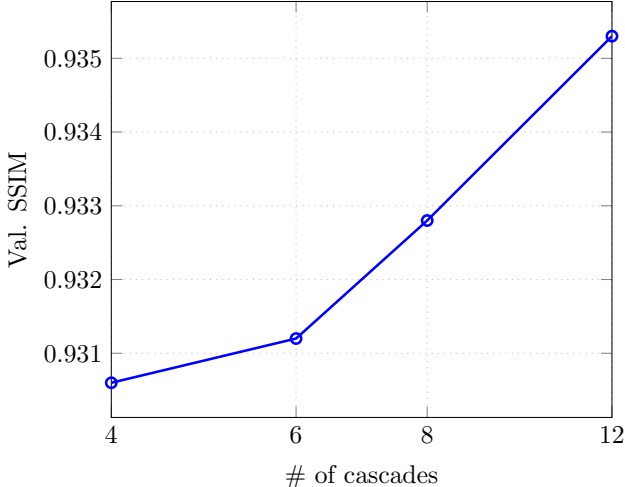

Figure 1: Validation SSIM as a function of number of cascades (unrolled itarations) in HUMUS-Net on the Stanford 3D dataset. We observe a steady increase in reconstruction performance with more cascades.

| Model | ASR? | SSIM | PSNR | NMSE |
|---|---|---|---|---|
| E2E-VarNet | ✗ | $0.9432 \pm 0.0063$ | $40.0 \pm 0.6$ | $0.0203 \pm 0.0006$ |
| E2E-VarNet | ✓ | $0.9457 \pm 0.0064$ | $40.4 \pm 0.7$ | $0.0186 \pm 0.0007$ |
| HUMUS-Net | ✗ | $\mathbf{0.9467 \pm 0.0059}$ | $\mathbf{40.6 \pm 0.6}$ | $\mathbf{0.0178 \pm 0.0005}$ |

Table 3: Results of the adjacent slice reconstruction ablation study on the Stanford 3D dataset. Mean and standard error over 3 random train-validation splits is shown. ASR improves the performance of E2E-VarNet. However, HUMUS-Net outperforms E2E-VarNet in all cases even without ASR.

## F    Iterative denoising visualization

Here, we provide more discussion on the iterative denoising interpretation of unrolled networks, such as HUMUS-Net. Consider the regularized inverse problem formulation of MRI reconstruction as

$$\hat{\boldsymbol{x}} = \arg\min_{\boldsymbol{x}} \left\| \mathcal{A}\left(\boldsymbol{x}\right) - \tilde{\boldsymbol{k}} \right\|^2 + \mathcal{R}(\boldsymbol{x}), \tag{F.1}$$

and design the architecture based on unrolling the gradient descent steps on the above problem. Applying data consistency corresponds to the gradient of the first loss term in (F.1), whereas we learn the gradient of the regularizer parameterized by the HUMUS-Block architecture yielding the update rule in k-space

$$\hat{\boldsymbol{k}}^{t+1} = \hat{\boldsymbol{k}}^t - \mu^t \boldsymbol{M}(\hat{\boldsymbol{k}}^t - \tilde{\boldsymbol{k}}) + \mathcal{G}(\hat{\boldsymbol{k}}^t).$$

This can be conceptualized as alternating between enforcing consistency with the measurements and applying a denoiser based on some prior, represented by the HUMUS-Block in our architecture.

We show an example of a sequence of intermediate reconstructions in order to support the denoising intuition in Figure 2. We plot the magnitude image of reconstructions at the outputs of consecutive HUMUS-Blocks, along with the model input zero-padded reconstruction. We point out that in general the intermediate reconstruction quality (for instance in terms of SSIM) is not necessarily an increasing function of cascades. This is due to the highly non-linear nature of the mapping represented by the neural network, which might not always be intuitive and interpretable.

## G    Vision Transformer terminology overview

Here we provide a brief overview of the terms related to the Transformer architecture, some of which has been carried over from the Natural Language Processing (NLP) literature. We introduce the key

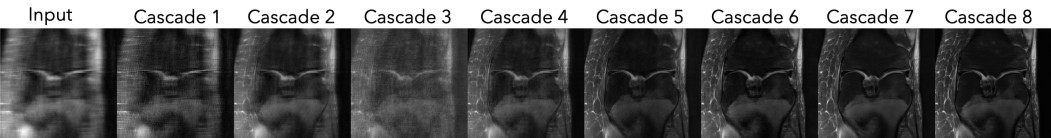

Figure 2: Visualization of intermediate reconstructions in HUMUS-Net.

concepts through the Vision Transformer architecture, which has inspired most Transformer-based architectures in computer vision.

Traditional Transformers for NLP receive a sequence of 1D token embeddings. Each token may represent for instance a word in a sentence. To extend this idea to 2D images, the input $x \in \mathbb{R}^{H \times W \times C}$ image ($H$ and $W$ stand for spatial dimensions, $C$ for channel dimension) is split into $N$ patches of size $P \times P$, and each patch is flattened into a 1D vector to produce an input of flattened patches $x_f$ with shape $N \times P^2C$:

$$x_f = \text{PatchFlatten}(x).$$

In order to set the latent dimension of the input entering the Vision Transformer, a trainable linear mapping $E \in \mathbb{R}^{P^2C \times D}$ is applied to the flattened patches, resulting in the so called patch embeddings $z$ of shape $N \times D$:

$$z = x_f E.$$

In order to encode information with respect to the position of image patches, a learnable positional embedding $E_{pos} \in \mathbb{R}^{N \times D}$ is added to the patch embeddings:

$$z_0 = z + E_{pos}.$$

The input to the Transformer encoder is this $N \times D$ representation, which we also refer to in the paper as token representation, as each row in the representation corresponds to a token (in our case an image patch) in the original input.

## H Detailed validation results

| Dataset | Model | SSIM(↑) | PSNR(↑) | NMSE(↓) |
|---|---|---|---|---|
| fastMRI | E2E-VarNet | 0.8908 | 36.8 | 0.0092 |
| | HUMUS-Net | **0.8934** | **37.0** | **0.0090** |
| Stanford 2D | E2E-VarNet | 0.8928 ± 0.0168 | **33.9 ± 0.7** | 0.0339 ± 0.0037 |
| | HUMUS-Net | **0.8954 ± 0.0136** | 33.7 ± 0.6 | **0.0337 ± 0.0024** |
| Stanford 3D | E2E-VarNet | 0.9432 ± 0.0063 | 40.0 ± 0.6 | 0.0203 ± 0.0006 |
| | HUMUS-Net | **0.9453 ± 0.0065** | **40.4 ± 0.6** | **0.0187 ± 0.0009** |

Table 4: Detailed validation results of HUMUS-Net on various datasets. For datasets with multiple train-validation split runs we show the mean and standard error of the runs.

## I Additional figures

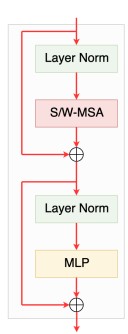

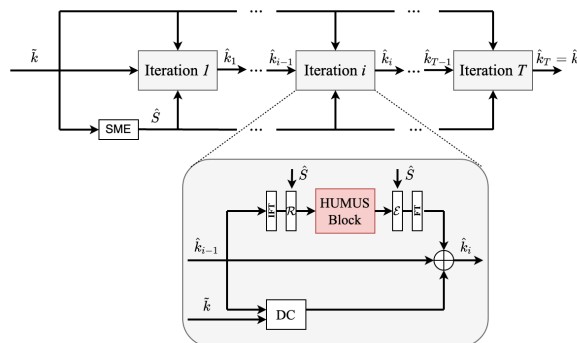

Figure 3: Swin Transformer Layer, the fundamental building block of the Residual Swin Transformer Block.

Figure 4: Depiction of iterative unrolling with sensitivity map estimator (SME). HUMUS-Net applies a highly efficient denoiser to progressively improve reconstructions in a cascade of sub-networks.

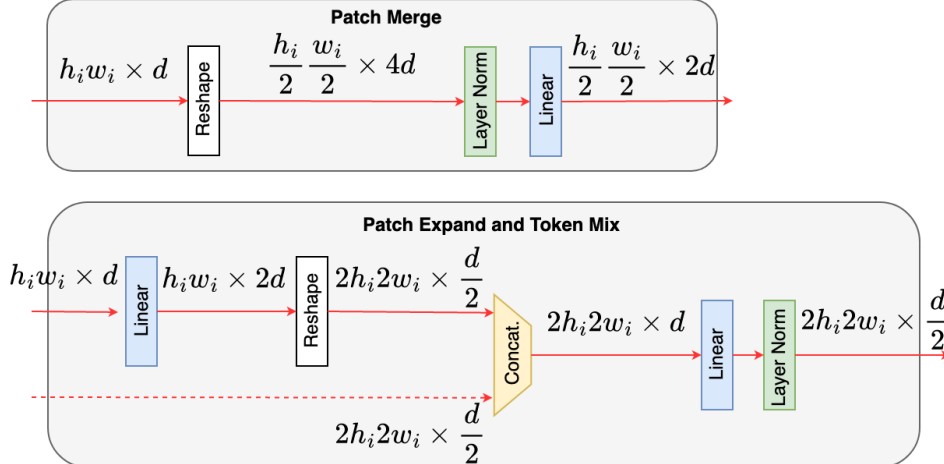

Figure 5: Patch merge and expand operations used in MUST.

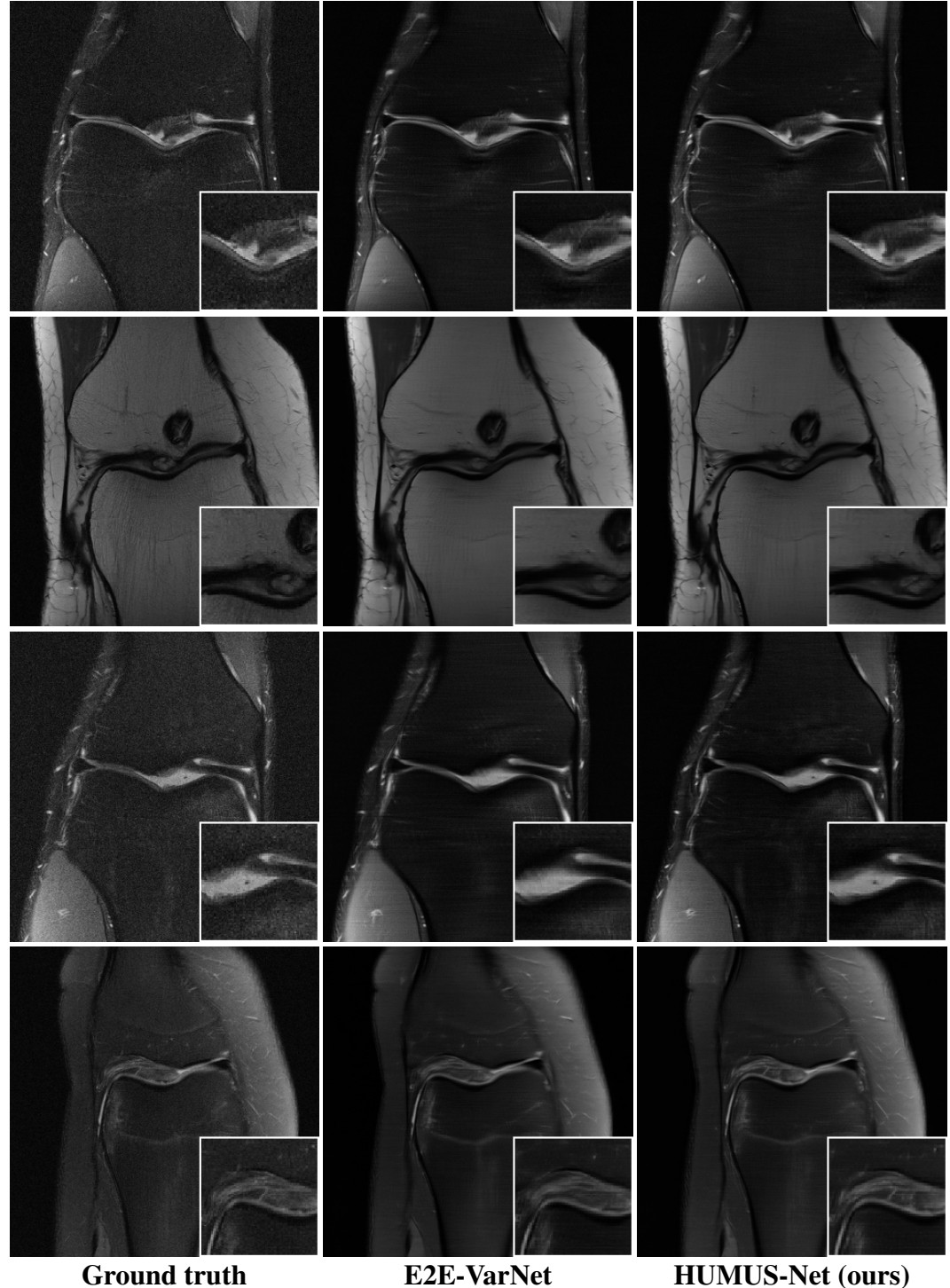

**Ground truth**       **E2E-VarNet**       **HUMUS-Net (ours)**

Figure 6: Visual comparison of reconstructions from the fastMRI knee dataset with $\times 8$ acceleration. HUMUS-Net reconstructs fine details on MRI images that other state-of-the-art methods may miss.