# OpenReview forum: "HUMUS-Net: Hybrid Unrolled Multi-scale Network Architecture for Accelerated MRI Reconstruction"
_NeurIPS.cc/2022/Conference — NeurIPS 2022 Accept_

### Official Review · Reviewer_ZMfj · 2022-07-10

**Rating:** 6
**Confidence:** 4
**Soundness:** 3 good
**Presentation:** 4 excellent
**Contribution:** 3 good

**Summary:**

This paper proposes a new state-of-the-art for accelerated MRI reconstruction. Different components of the model are properly explained by the authors and most of the design choices are well-justified. There are also several concerns/comments pointed out below.

**Questions:**

The reviewer understands the point with long-range-dependancy modeling of Transformers. However, it is unclear why that should be beneficial in reconstruction tasks such as MRI reconstruction. Note that locality matters for image reconstruction which is captured by convolutions.

While the ablation studies are interesting, two important parts are missing: (1) Adjacent Slice Reconstruction (ASR), and (2) Residual learning. It would be interesting to see how the E2E-VarNet performs if these two elements are added. This is important since if with these modifications, the performance of E2E-VarNet matches HUMUS-Net, then **it sends the message that a complicated HUMUS-Block is not necessary and one can simply regularize the unrolled problem with a simple U-Net (as E2E-VarNet currently does)**.

Given the marginal improvement of HUMUS-Net over E2E-VarNet, what are some competitive advantages of HUMUS-Net that could convince someone to use it instead of E2E-VarNet? e.g., is it computationally cheaper to train? Does it require less training data? Is it more robust? etc. Any explanations in this direction are highly appreciated.

It would be nice to add the number of parameters of each model to Table 2 to assure the reader of the fairness of the comparisons.

Typos

- Line 193: a significant challene=
- Line 199: In this work,

**Limitations:**

The reviewer can discuss limitations of the work better if the authors answer the questions/concerns raised above in the Questions/Suggestions section.

**Strengths And Weaknesses:**

++ The paper is very well-written and easy to follow. Having a new state-of-the-art for accelerated MRI reconstruction is also highly appreciated as it has not been pushed since 2020.

++ Most of the architectural design choices are properly justified by the authors.

+- The proposed method is very intuitive in most of its aspects (e.g., high- and low-res processing paradigms). However, some explanations are vague, please see the comment below in the questions/suggestions part.

-- Ablation studies are not conclusive (please see the comment below in the questions/suggestions part).

-- Evaluation experiments can be improved w.r.t efficiency of the model and fairness of the comparisons.

---

> ### Author Response · Authors · 2022-08-02
> **Response to Reviewer pt. 1**
>
> We really appreciate the reviewer’s insightful comments and suggestions. We are glad that the reviewer also thinks that “having a new state-of-the-art for accelerated MRI reconstruction is also highly appreciated as it has not been pushed since 2020”. Please find our response to the reviewer’s comments below.
>
> Re importance of long-range dependency modeling: We agree with the reviewer that local spatial correlations are crucial for reconstruction problems, and the success of fully convolutional models clearly shows that high quality reconstructions can be obtained relying mostly on locality. However, the deeper convolutional networks get, the larger their receptive field becomes and therefore these networks are also capable of learning and utilizing long-range dependencies. The benefit of Transformer architectures stems from their built-in ease of modeling such dependencies in the form of self-attention. That being said, we believe that having access to long-range pixel dependencies can help in reconstruction problems. For instance, certain diseases may manifest in multiple small, characteristic features scattered across the imaged anatomy. In such cases, directly correlating such features may be more efficient than attempting to individually and locally recovering them.
>
> Re extra ablation studies: Thanks, this is a great suggestion and indeed worth including in the paper. We observed that even though adding adjacent slice reconstruction to VarNet typically helps, 1) HUMUS-Net significantly outperforms VarNet even without adjacent slice reconstruction and 2) VarNet with adjacent slice reconstruction is still outperformed by HUMUS-Net. To highlight 1), we emphasize that HUMUS-Net used a single slice reconstruction for all Stanford2D and Stanford3D experiments, and still obtained consistently higher SSIM than VarNet. To demonstrate 2), we added a new set of experiments in Appendix D on VarNet enhanced with 3 slice reconstruction. We observe that HUMUS-Net with single slice reconstruction outperforms VarNet with 3 slice reconstruction. We agree with the reviewer that residual learning can add some extra gains to VarNet, but we don’t believe that it is significant enough to approach HUMUS-Net performance. We made an honest attempt at adding a skip connection by-passing the U-Net denoiser in E2E-VarNet on the Stanford 3D dataset, however in our experiments, it performed worse than vanilla VarNet. We would like to point out that adding skip connections can fundamentally change the dynamics of model training (information and gradient flow in the network) and such it may require significant further modifications to VarNet and extensive hyperparameter tuning which is outside of the scope of our current paper. We hope that we were able to convince the reviewer that, even though adjacent slice reconstruction is a nice addition to any architecture, it is not the sole reason why HUMUS-Net performs so well.
>
> Re advantages of HUMUS-Net: Thank you for the question, let us elaborate more on the competitive advantage of HUMUS-Net. First, as we have explained to another reviewer, the improvement in SSIM is indeed significant. In the past years, much effort has been made in designing MRI reconstruction architectures that perform well on a large-scale dataset such as fastMRI. The fully-convolutional models tend to saturate around SSIMs close to VarNet’s performance and no significant improvement has been published on this dataset in the past two years (see 0.8900 SSIM VarNet vs 0.8893 SSIM XPDNet the second best) and therefore we believe that HUMUS-Net’s improvement is significant on this extremely competitive benchmark dataset. From a practical standpoint, we included sample reconstructions in Appendix H from HUMUS-Net and VarNet demonstrating that there are details lost in VarNet reconstructions that can be recovered by HUMUS-Net. This has potential merit in medical diagnosis as well, as diagnosis of certain diseases relies on very fine features on the reconstructions. When it comes to data-efficiency, our network trained on *only* the training split beats the previous best E2E-VarNet that has been trained on 20% more data (both training and validation split) by a margin. We agree with the reviewer that it would be a very exciting future direction to look into the robustness of HUMUS-Net and Transformer-based MRI reconstruction methods in general. As we have mentioned for another reviewer, the combination of HUMUS-Net with recent results in test-time training [1] could have great potential in robust MRI reconstruction.
>
> Please find below the second part of our response.

---

> > ### Author Response · Authors · 2022-08-02
> > **Response to Reviewer pt. 2**
> >
> > Re number of parameters: Thanks for pointing this out. For the models in Table 1, HUMUS-Net has 109M parameters, E2E-VarNet has 30M, XPDNet has 155M and U-Net has 214M. We were unable to find the exact number of parameters in the other models’ corresponding papers, but have reached out to the authors and will update the reviewer with the numbers and include them in the final version of the paper.
> >
> > Re typos: Thank you, we fixed the typos.
> >
> > [1] Darestani, Mohammad Zalbagi, Jiayu Liu, and Reinhard Heckel. "Test-Time Training Can Close the Natural Distribution Shift Performance Gap in Deep Learning Based Compressed Sensing." arXiv preprint arXiv:2204.07204 (2022).

---

> > > ### Comment · Reviewer_ZMfj · 2022-08-08
> > > **Response to authors**
> > >
> > > The reviewer would like to first thank the authors for their great efforts in both revising the manuscript and providing a conclusive response. The experiments are more conclusive now. Aside from all the properly-addressed points by the authors, the reviewer would like to discuss a few points further.
> > >
> > > It is indeed a valid point that the fastMRI dataset is very competitive and the authors are right that many efforts have failed to push the state-of-the-art over the past two years. That is why the 0.0030 higher SSIM achieved by HUMUS-Net is appreciated.
> > >
> > > However, what would derive the significance of the contribution of this paper is at least a comparison between VarNet and HUMUS-Net with equal number of parameters. Currently, we have to speculate that HUMUS-Net will outperform VarNet with an equal model size (the case with U-Net is different for instance because the results show for a fact that a large U-Net is outperformed by a smaller HUMUS-Net).
> > >
> > > The reviewer understands that it is unreasonable to ask for this experiment now given the timeline. But note that adjacent slice reconstruction reduces the gap between VarNet and HUMUS-Net significantly, so why wouldn't increasing VarNet's model size?
> > >
> > > The reviewer also isn't fully convinced by the competitive advantages of HUMUS-Net. It seems VarNet, which is regularized by a simple U-Net, achieves almost the same performance as HUMUS-Net but with a much more simpler architecture; this is a competitive advantage.
> > >
> > > Minor point:
> > > How come Table 5 says:
> > >
> > > `VarNet               0.9432
> > > HUMUS-Net     0.9467`
> > >
> > > but Table 3 shows:
> > >
> > > `VarNet               0.9432
> > > HUMUS-Net     0.9449`
> > >
> > > and Table 2 shows:
> > >
> > > `VarNet               0.9432
> > > HUMUS-Net     0.9453`
> > >
> > > Do these belong to different runs of HUMUS-Net?

---

> > > > ### Author Response · Authors · 2022-08-09
> > > > **Response to Reviewer**
> > > >
> > > > Thank you for putting in the extra effort going over our rebuttal and we appreciate the further comments. We are glad that we were able to address most of the reviewer’s questions and suggestions. Please find below our response to the remaining comments.
> > > >
> > > > First, we would like to point out that a higher number of parameters doesn’t necessarily correlate with better performance. With updated numbers from all authors that have replied to us so far:
> > > > - HUMUS-Net:109M parameters
> > > > - E2E-VarNet: 30M
> > > > - XPDNet: 155M
> > > > - U-Net: 214M
> > > > - i-RIM: approx. 300M
> > > >
> > > > That said, we agree with the reviewer that it would be an interesting direct comparison with VarNet to try to match the number of parameters. To this end, we designed a tiny version of HUMUS-Net, we call it *HUMUS-Net-tiny*, with only 26M parameters by reducing the number of cascades to 6 and the embedding dimension to 36. Furthermore, we *removed adjacent slice reconstruction and residual learning* from HUMUS-Net in order to provide the most direct comparison. Unfortunately, the time frame was too short to complete the above experiments, but the preliminary results are very promising. We are going to include large-scale experiments in the camera-ready version of the paper in addition to such smaller scale comparisons.
> > > >
> > > > We would also like to point out that adjacent slice reconstruction improved E2E-VarNet’s performance in our ablation study, but it still performed worse than the HUMUS-Net model *without* ASR.
> > > >
> > > > With respect to the reviewer’s comment on competitive advantage, we agree that E2E-VarNet’s architecture is simpler. However, our work has demonstrated that in order to surpass the current state-of-the-art, that has been unchanged in the past years, novel ideas are required, such as our hybrid Transformer-convolutional architecture.  We agree with the reviewer that in some applications, the reconstruction quality obtained from E2E-VarNet is sufficient, and in fact this can be true for most other MRI reconstruction architectures closely following E2E-VarNet on the leaderboard. However, when the best reconstruction quality is required, currently HUMUS-Net has the competitive advantage.
> > > >
> > > > Re minor point: Table 2 result is on the default HUMUS-Net architecture, Table 3 result is on a smaller HUMUS-Net architecture we designed for fairness of comparison in the ablation study (embedding dimension reduced to 36 as described in Section 5.2.), and Table 5 is the result on HUMUS-Net where we matched the number of cascades with E2E-VarNet. We are going to make this more clear in the final version, thank you for pointing this out.

---

> > > > > ### Comment · Reviewer_ZMfj · 2022-08-09
> > > > > **Response to authors**
> > > > >
> > > > > Thanks for the reply.
> > > > >
> > > > > It is true that more parameters may not necessarily translate into matching performance as we see in the comparison between U-Net and HUMUS-Net, but note that within-model correlation typically exists. i.e., a U-Net with 100 M parameters outperforms a U-Net with 10 M parameters. That's why one can expect to see a better performance from a VarNet with 109 M parameters than a VarNet with 30 M.
> > > > >
> > > > > Thanks for running the new experiment, although I did not expect this in such short amount of time.
> > > > >
> > > > > Finally, I still find the argument with the competitive advantage not convincing.
> > > > >
> > > > > To conclude this discussion, I find a few points not convincing or not properly justified as mentioned in my review and responses to the authors. That being said, I'm very satisfied with the rebuttal and believe that it has surely added more value to the paper. Thus, it is reasonable to adjust my score and I'll raise it to `6`.
> > > > > The reviewer would like to thank the authors again for their time and effort.

---

### Official Review · Reviewer_pRk7 · 2022-07-11

**Rating:** 5
**Confidence:** 4
**Soundness:** 3 good
**Presentation:** 3 good
**Contribution:** 2 fair

**Summary:**

The paper describes a method for accelerated parallel MRI reconstruction using deep neural networks. This is inspired by recent advancements of vision transformer (ViT) and deep unfolding (also known as algorithm unrolling) network for medical imaging. In specific, a new network architecture (including convolutional and self-attention layers) was proposed and used as a denoising prior in the gradient descent unrolling iterations. The proposed reconstruction network, named as HUMUS-Net, is then trained in an end-to-end supervised fashion, where ground truth images are used during training. The authors compare their proposal against existing SOTAs such as U-Net and the End-to-End Variational network (E2E-VarNet) using the fastMRI and Stanford 2D/3D datasets. Numerical results show that HUMUS-Net achieves comparable or slightly better results to E2E-VarNet and significantly outperforms traditional U-Net.

**Questions:**

1), The HUMUS-Net models reported in Table 1 do not show a large improvement compared to E2E-VarNet in terms of PSNR and NMSE. This especially true for when models trained on the fastMRI using training dataset only. Since the proposal in this work uses exactly the same unrolling algorithm with jointly coil estimation, it is unclear the rational for using sophisticated transformer as denoising priors, compared to the simple U-Net used in E2E-VarNet.

2), As mentioned by the authors, The MUST module in this work is inspired by the residual Swin transformer blocks. Consider explaining and discussing the relations and differences between the proposal MUST and Swin transformer in more details. A deeper understanding and intuition will strengthen this work and make its contribution more solid.

3), In abstract it is claimed that the HUMUS-Net consume less memory compared to existing transformer-based methods. The memory consumption of proposal should scale with number of iterations. Will unroll more iterations results in larger memory usage? In addition, are the weights of HUMUS-Net shared cross each iteration? If not, this will also affect the disk storage size.

4), The ablation studies in Sec. 5.2, comparisons to SwinIR transformer denoiser seems to be unfair and not very informative. As mentioned by the authors, SwinIR was not originally designed for fMRI reconstruction. Using existing transformers for fMRI will be more precise. In addition, I do not think the simple SwinIR (reduced embedding dimension due to lack of gpu memory) is fine-tuned as much as HUMUS-Net in this work. Maybe the authors can use memory efficient training such as checkpoint or deep equilibrium model (DEQ), while keeping the model’s structure as in its original work.

5), As mentioned earlier, some important numerical results, ablation studies and technical details are missing.

    (a) What is training loss used in this work?
    (b) PSNR values are missing in Table 2. and Table 3.
    (c) Visual comparisons of the proposal to baseline methods are missing in the main manuscript.
    (d) Different acceleration ratios (e.g., X4, X6, etc.) to E2E-VarNet are missing.
    (e) Number of parameters and running time comparisons to baseline and other transformer-based fMRI methods, such as [R1, R2] are missing.

6), While this may be orthogonal to this proposal, it would be helpful to discuss the robustness of HUMUS-Net to model shifting (e.g., anatomy shift etc.) and how is the HUMUS-Net generalizable to other medical imaging modalities. These will strength the current paper.

Ref:
[R1] Lin. et al., “Vision Transformers Enable Fast and Robust Accelerated MRI”, MIDL, 2022.
(Github: https://github.com/MLI-lab/transformers_for_imaging)

[R2] Korkmaz. et al., “Unsupervised MRI Reconstruction via Zero-Shot Learned Adversarial Transformers”, IEEE TMI, 2022.
(Github: https: //github.com/icon-lab/SLATER.)


**Limitations:**

The limitations of this proposal are missing and need to be discussed in the revision.

**Strengths And Weaknesses:**

This paper is overall well written and reports competitive results with both quantitative and qualitative validation for fMRI reconstruction. However, the methodological contributions and rationale for using transformer in unrolling are unclear or at least not well illustrated in its current state. The supportive numerical ablation studies (e.g., running time, memory usage, robustness to model shifting, comparison to existing transformer-based fMRI methods) are missing. Please find my technical comments bellow.

---

> ### Author Response · Authors · 2022-08-02
> **Response to Reviewer pt 1**
>
> We thank the reviewer for the in-depth comments and suggestions on how to further strengthen our paper. Please see below our response to the points raised.
>
> Re 1.: The reviewer raises concern about the magnitude of improvement HUMUS-Net provides over E2E-VarNet. We would argue that the improvement is indeed significant. In the past years,  much effort has been made in designing MRI reconstruction architectures that perform well on a large-scale dataset such as fastMRI. The fully-convolutional models tend to saturate around SSIMs close to VarNet’s performance and no significant improvement has been demonstrated in the literature on this dataset in the past two years (see 0.8900 SSIM VarNet vs 0.8893 SSIM XPDNet the second best). Our network trained on *only* the training split  beats the previous best E2E-VarNet that has been trained on 20% more data (both training and validation split) by a margin. We believe this is significant. We included sample reconstructions in Appendix H from HUMUS-Net and VarNet demonstrating that there are details lost in VarNet reconstructions that can be recovered by HUMUS-Net. This has potential merit in medical diagnosis as well, as diagnosis of certain diseases relies on very fine features on the reconstructions. Finally, the improvement is most significant in terms of SSIM, as this is the loss we optimized HUMUS-Net over. We expect improvement in other metrics using a combined loss function. Furthermore, we would like to point out that NMSE is not considered a good reconstruction metric by many, since it prefers overly smooth reconstructions. We hope that our response has addressed the reviewer’s concern about the significance of the improvement provided by HUMUS-Net, and would be happy to receive further specific comments in this direction that we might be able to further clarify/address.
>
> Re 2.: Thank you for the suggestion. We discussed what design choices and considerations led us from the SwinIR architecture to MUST and the HUMUS-Block in Section 5.2. In Section 4.2. we discuss how we modified the fundamental building block of SwinIR, the Residual Swin Transformer Block, in order to create a multi-scale architecture. If the reviewer has some concrete suggestion in which direction we could expand on the comparison, we would gladly include extra discussion in the main paper.
>
> Re 3.: First, we would like to clarify that we do not state in the abstract that HUMUS-Net consumes less memory than other Transformer-based methods in general. In fact, at the time of writing this paper there was no other Transformer-based architecture designed for supervised accelerated MRI reconstruction that is competitive on fastMRI and the required input resolutions. In the abstract we argue that existing Transformer architectures cannot handle dense problems where the input resolution is high, and a large number of patches are required for good performance. Existing Transformer architectures have excellent performance on classification problems, where fine details are not too important for accuracy, and medical segmentation tasks where we can get away with a much smaller number of image patches for good performance; thus the memory barrier doesn’t necessarily arise in these applications. The few existing Transformer-based approaches to MRI reconstruction are not competitive on large benchmark datasets. That being said, the reviewer is correct that memory consumption scales linearly with the number of unrolled iterations, which limits network depth. We now provided ablation studies how HUMUS-Net scales with the number of cascades in Appendix C, we invite the reviewer to take a look for more information. Finally, the weights are not shared between different cascades, which indeed increases storage requirements. This way different cascades can learn to implement different forms of denoisers. It would be interesting to see how the network performs with shared weights across cascades.
>
> Please find the second part of our response below.

---

> > ### Author Response · Authors · 2022-08-02
> > **Response to Reviewer pt. 2**
> >
> > Re 4.: The reviewer raises concerns about the ablation studies in Section 5.2. as they find it not very informative. We believe there is some misunderstanding with respect to the goal of these ablation studies. In Section 5.2, we would like to guide the reader through what design choices and steps lead us from the SwinIR architecture to HUMUS-Net. We agree with the reviewer that SwinIR is not a fair comparison for HUMUS-Net, as it has been designed for a fundamentally different problem, where the specific considerations of MRI reconstruction do not arise. However, our goal is not to make direct comparison between these architectures, but to explain the weaknesses of SwinIR for MRI reconstruction and show how we addressed those to arrive at the HUMUS-Block architecture. We demonstrate step-by-step how each design choice improves the performance on an MRI reconstruction problem. We hope this clarifies the goal of this section. We now modified the text in the main paper to better reflect this goal.
> >
> > Re 5. a): Thank you for pointing this out, we now added the loss to Appendix A. We optimize the SSIM loss.
> >
> > Re 5. b): We now added the table showing detailed results in Appendix G.
> >
> > Re 5. c): Due to the space limitation in the main paper, we included high resolution visual comparison of reconstructions in Appendix H. We believe that squeezing lower resolution plots in the main paper would make it difficult to see the fine differences in reconstructions.
> >
> > Re 5. d): Thank you for the suggestion. We focused on 8x acceleration as it is the more challenging problem when compared with lower accelerations. As we have mentioned in the response to another reviewer, we have some very promising indicators that HUMUS-Net would perform well on 4x acceleration as well. HUMUS-Net trained on x8 acceleration only on training data achieves 0.9238 SSIM on the x4 acceleration test dataset as seen on the leaderboard, which is higher than the value of 0.9100 SSIM reported for E2E-VarNet x4 on the validation dataset in [1] (Table 2).
> >
> > Re 5. e): Thank you for the suggestion. As we explain to another reviewer, for the models in Table 1, HUMUS-Net has 109M parameters, E2E-VarNet has 30M, XPDNet has 155M and U-Net has 214M. We were unable to find the exact number of parameters in the other models’ corresponding papers, but have reached out to the authors and will update the reviewer with the numbers and include them in the final version of the paper. We are aware of the interesting work of [R1] and discuss it in our paper. However, [R1] can be considered as a proof-of-concept for fully Transformer-based methods in MRI reconstruction, whereas HUMUS-Net is a hybrid network. [R1] focuses more on low data training, robustness consideration and pre-training. Their model is not competitive on the fastMRI dataset as it performs worse than VarNet. Thank you for bringing our attention to [R2], we have now added it to the Related Work section. However, we don’t think this technique is comparable to our algorithm, because it is a 1) unsupervised method, whereas HUMUS-Net is fully supervised and 2) it is not an end-to-end reconstruction technique. Due to the fact that it is unsupervised, we cannot expect comparable performance to strong supervised baselines.
> >
> > Re 6.: This is a great suggestion and would indeed be a very interesting direction to explore. The combination of HUMUS-Net with recent results in test-time training [2] could have great potential in robust MRI reconstruction. Performing careful studies on the robustness of hybrid and Transformer-based reconstruction techniques compared with fully convolutional methods would be an exciting future direction.
> >
> > [1] Sriram, Anuroop, et al. "End-to-end variational networks for accelerated MRI reconstruction." International Conference on Medical Image Computing and Computer-Assisted Intervention. Springer, Cham, 2020.
> >
> > [2] Darestani, Mohammad Zalbagi, Jiayu Liu, and Reinhard Heckel. "Test-Time Training Can Close the Natural Distribution Shift Performance Gap in Deep Learning Based Compressed Sensing." arXiv preprint arXiv:2204.07204 (2022).

---

> > > ### Comment · Reviewer_pRk7 · 2022-08-08
> > > **Thank you**
> > >
> > > Thank the authors for the detailed replies! It resolves most of my confusions. After considering other reviewer comments during the rebuttal and my intial understanding, however, the reviewer still cannot fully agree with the usage of transformers in deep unrolling due to the incremental improvements over VarNet. The proposed method seems to be too specific for MRI and the broaden impacts to NeurIPS community are unclear.
> > >
> > > As a result, I think this work is on the borderline of this year's NeurIPS and I would like to maintain my borderline acceptance.

---

> > > > ### Author Response · Authors · 2022-08-09
> > > > **Response to reviewer**
> > > >
> > > > Thank you for your response and for going over our answers. We are happy to hear that we have resolved most of your issues and confusion with our paper.
> > > >
> > > > As we have discussed in our rebuttal and in response to other reviewers, we believe that the improvement provided by HUMUS-Net over E2E-VarNet is indeed significant. The fastMRI leaderboard is extremely competitive, and the state-of-the-art has not been pushed in years. Our network beats the current state of the art, but with 20% less training data, which is not an incremental improvement in our opinion.
> > > >
> > > > While we agree with the reviewer that our method is specific to medical imaging, we do believe the techniques may be useful more broadly in other reconstruction/regression tasks including in other scientific imaging domains. That said, we would like to note that as modern AI is increasingly used for healthcare applications, understanding the performance of modern deep learning on medical imaging and developing architectures and methods well suited to this setting is rather important. We emphasize that MRI is a very important imaging modality of significant diagnostic import in various medical domains and improvements in its speed, cost, accuracy etc can thus lead to many societal benefits. We thus think it is relevant to the NeurIPS community. Indeed, the NeurIPS community has shown great interest in this area as evident by the annual “Medical Imaging Meets NeurIPS” workshops in 2020, 2021, and 2022, where several talks are specifically on MRI reconstruction methods on the competitive fastMRI dataset.

---

### Official Review · Reviewer_4jKX · 2022-07-11

**Rating:** 7
**Confidence:** 3
**Soundness:** 3 good
**Presentation:** 3 good
**Contribution:** 4 excellent

**Summary:**

The paper proposes HUMUS-Net, a NN architecture that combines convolutions and transformer blocks for compressed sensing MRI reconstruction. The approach achieves SOTA on the benchmark fastMRI dataset with 8x acceleration. The code is available with the supplementary materials.

**Questions:**

1. The experiments are done on 8x acceleration. It seems that this approach might also be fitting well for further acceleration ratios >8x. Did you consider validating the method on higher acceleration ratios and if not, how do you expect the method to perform? How about lower acceleration, 4x?
2. You mention that the current architecture requires fixed-size inputs. Could you explain why? Both extractor and reconstruction blocks are convolutional, so it shouldn't be the problem.

**Limitations:**

The authors adequately addressed the limitations and potential negative societal impact of their work.

**Strengths And Weaknesses:**

Strengths:
1. The novel approach paves way for applied MRI models with hybrid convolutional-transformer architectures.
2. The paper carefully describes design decisions and includes ablation study of the model.

Weaknesses: -

---

> ### Author Response · Authors · 2022-08-02
> **Response to Reviewer**
>
> We appreciate the reviewer’s effort on our paper and giving us valuable feedback. We are especially glad to hear that our “novel approach paves way for applied MRI models with hybrid convolutional-transformer architectures”, which was exactly our main goal. Please see below our response to your questions.
>
> Re 8x acceleration: We agree with the reviewer that it would be indeed very interesting seeing HUMUS-Net reconstructions on higher acceleration ratios, such as x10 or x16 and we see no reason why our proposed model would not perform well in that regime. We have chosen 8x acceleration ratio as this is the more challenging problem, which also has baselines on most datasets. When it comes to lower acceleration ratios, we have some very promising indicators that HUMUS-Net would perform well. In fact, our model trained on x8 acceleration only on training data achieves 0.9238 SSIM on the x4 acceleration test dataset as seen on the leaderboard, which is higher than the value of 0.9100 SSIM reported for E2E-VarNet x4 on the validation dataset in [1] (Table 2).
>
> Re fixed size inputs: Thank you for the question. Our MUST encoder-decoder architecture utilizes SwinTransformer layers, that rely on fixed window sizes within which the self-attention is computed. Thus the input size has to be divisible by the fixed window size (typically 7x7 or 8x8 patches). One approach to address changing input shapes would be to dynamically adjust the window size based on input shape, however this would require a careful study as it can have unexpected consequences. Another way would be to use rectangular window shapes, which could also lead to more interesting discussion.
>
> [1] Sriram, Anuroop, et al. "End-to-end variational networks for accelerated MRI reconstruction." International Conference on Medical Image Computing and Computer-Assisted Intervention. Springer, Cham, 2020.

---

> > ### Comment · Reviewer_4jKX · 2022-08-10
> > **Thanks!**
> >
> > Thank you for answering reviewers’ questions and addressing the feedback. I stand by my initial rating and consider this paper a nice contribution to the community.

---

### Official Review · Reviewer_on3J · 2022-07-12

**Rating:** 5
**Confidence:** 4
**Soundness:** 3 good
**Presentation:** 3 good
**Contribution:** 3 good

**Summary:**

This paper proposed the HUMUS-Net that used to achieve the goal of reconstructing accelerated  multi-coil MRI images. HUMUS-Net consists several cascades of HUMUL-Blocks, each block complete a step in unrolled iterations of an optimization algorithm to solve inverse problem. Each HUMUL-Block contains three blocks: high-resolution feature extraction, low-resolution feature extraction and high-resolution image reconstruction. Further, the core component of HUMUL-Block was specified. It is a multi-scale hybrid feature extraction component named MUlti-scale residual Swin Transformer network (MUST). It adopted the Residual Swin Transformer Blocks (RSTB) in a hierarchical encoder-decoder architecture to facilitate the multi-scale processing. The MUST is comprised several blocks of RSTB-B, RSTB-D and RSTB-U.  The usage of HUMUL-Block dedicate to solve the regularization term in image domain was indicated. At last, the benefit of reconstruct image using adjacent slices to remove artifacts were explained.
Performance of several state-of-the-art reconstruction techniques on the fastMRI knee multi-coil ×8 dataset were compared with respect to metrics of SSIM, PSNR and NMSE. HUMUS-Net and E2E-VarNet were validated as well on three different datasets.  And the ablation studies have done to justify the proposed design.


**Questions:**

Detailed comments:
1. The model split input images into patches, and embed the patches into lower-dimensional tokens. The way to produce tokens is not clear.
2. Line 180-182: As shown in Figure 6 in supplementary, HUMUL-Block only dedicates to the regularization term in (Eq 2.2). The high-level unrolled architecture is based on the k-space domain. The expression on line 180-182 seems tell us that each cascade works on the image domain, which is confusing.
3. Line 192: The used resolution of MRI image is not indicated, makes it hard to compare with (32x32-256x256).
4. Line 216-218: The explanation of “token representation”, “learned linear mapping” and “positional encoding” are missing.
5. Line 219: what does the phrase of 'Tokens corresponding to different image patches....in a concise latent representation' means? It is too abstract for me.
6. Figure 2 and 3: The font of the dimension indicator is too small.
7. Figure 3: The reshape blocks are not explained.
8. Line 268: Please specify the dimension of concatenated k-space slices and the dimension of input into the HUMUL-Block
9. Line 329: How does 10 times was calculated?


**Ethics Review Area:**

["I don’t know"]

**Limitations:**

some ablation study is not provided.

**Strengths And Weaknesses:**

Strength：
The proposed HUMUS-Net claimed to have several strengths:
1. The utilization of self-attention based Transformer can overcome the weakness of convolutions, like content independence and the inability to model long-range dependencies.
2. The multi-scale structure in its core block MUST not only could reconstruct fine details but also reduced the computational cost.
3. The model has achieved the second position in fastMRI leaderboard of ×8 multi-coil knee challenge.
4. The proposed method performs good at three various dataset: fastMRI knee dataset, Stanford2D and Stanford3D dataset.

Weakness：
1. The proposed HUMUS-Net achieved better results than the other methods with respect to metrics of SSIM, PSNR and NMSE. But the radiologist's evaluation on specific images is missing.
2. The HUMUL-Block in HUMUS-Net claimed acts like an intermediate denoiser. But the analysis of this effect is missing.
3. The ablation experiment of cascade with respect to reconstruction performance is missing.

---

> ### Author Response · Authors · 2022-08-02
> **Response to Reviewer pt. 1**
>
> We thank the reviewer for their insight on our paper and appreciate the detailed comments. We are glad that the reviewer has an overall positive opinion of our work. We address the raised issues below.
>
> Re Weakness 1.: The reviewer raises a very good point in that a radiologist’s evaluation of the reconstructions would be extremely valuable and would provide an extra dimension to compare the algorithms. We are in fact working on setting up a board of medical professionals from University of Southern California, University of Utah, University of Washington and NYU for ranking HUMUS reconstructions. We strongly believe that the higher quality reconstructions provided by HUMUS-Net can make a difference in medical diagnosis. We invite the reviewer to take a look at some high-resolution sample reconstructions in Appendix H, demonstrating that in some cases HUMUS-Net can reconstruct fine details on the anatomy that has been missed by other methods, such as E2E-VarNet.
>
> Re Weakness 2.: We consider the regularized inverse problem formulation of MRI reconstruction (Eq. 2.1) and design the architecture based on unrolling the gradient descent steps on the above problem. Applying data consistency corresponds to the gradient of the first term in Eq. 2.2, whereas we learn the gradient of the regularizer parameterized by the HUMUS-Block architecture. This can be conceptualized as alternating between enforcing consistency with the measurements and applying a denoiser based on some prior (represented by the HUMUS-Block). We have now added a discussion and visualization of this process through a sample reconstruction as it passes through the cascades of the network in Appendix E. We plan to move some of this discussion to the main paper in the final version if we get an extra content page. Please let us know if you would like to see something specific.
>
> Re Weakness 3.: This is indeed a valuable suggestion. In finding the optimal number of cascades we performed a smaller scale ablation study on the Stanford 3D dataset. We observed that higher number of cascades perform better in general. Therefore, the bottleneck in choosing the number of cascades has been the GPU memory limitation. We have now attached an ablation study on the number of HUMUS-Net cascades in Appendix C. These experiments suggest that with access to more GPU memory, HUMUS-Net could potentially even surpass the current results by increasing the number of unrolled iterations.
>
> Detailed comments 1.: We produce tokens by flattening the patches along the spatial dimensions followed by a learned linear transformation along the channel dimension, similar to how it is done traditionally for Vision Transformers. However, we observed no positive impact on performance in using the learned linear projections, so we omit that. We further clarify this in the main paper.
>
> Detailed comments 2.: Thanks for pointing this out. The reviewer is right that the high level unrolling is in k-space domain, as seen in Eq. 2.2. However, the HUMUS-Block acts on the image domain representation after inverse Fourier-transforming the current k-space reconstruction as seen in line 254. We now better highlight this in the main paper.
>
> Detailed comments 3.: Good point. HUMUS-Net inputs have 384x384 spatial resolution in the experiments, after center cropping and padding. We indicated this in Appendix A, but now also moved it to the main paper to better highlight the difference.
>
> Detailed comments 4.: Thank you for the comment, we now added more details in Appendix F to clarify the terminology.
>
> Detailed comments 5.: Let us clarify this sentence. In the encoder path, we repeatedly merge tokens and expand their embedding
> dimension, thus as we go deeper in the network we have less and less tokens with higher and higher dimensional representation. This is analogous to U-Net, where strided convolutions are used to reduce spatial dimension and increase the channel dimension. The final representation in the bottleneck consists of a very low number of tokens, each of which contains a “summary” of the encoded information and is often referred to as the latent space of the encoder in autoencoder literature.
>
> Detailed comments 6.: Thanks, we updated the image for better readability.
>
> Detailed comments 7.: Thank you for pointing this out. The reshape blocks either concatenate (PatchMerge) or split (PatchExpand) tokens along the embedding dimension, which is analogous to convolutional up- or downsampling in U-Nets. The ‘learned’ part of the above transformations come from a linear projection after the reshape operation. We have added now a more detailed figure showing all intermediate feature shapes in Appendix H.
>
> Please see the second part of our response below.

---

> > ### Author Response · Authors · 2022-08-02
> > **Response to Reviewer pt. 2**
> >
> > Detailed comments 8.: The concatenated k-space slices have shape $2 \cdot s \cdot c \times h \times w$, where $c$ is the number of coil channels, $s$ is the number of reconstructed slices ($s=3$ for adjacent slice reconstruction of 3 slices, 1 in the vanilla case) and the multiplication by 2 represents the real and imaginary parts; and $h$ and $w$ stand for the image dimensions. In other words, the slices are stacked along the coil dimension with real part and imaginary parts as separate channels. The input to the HUMUS-Block is the current multi-coil k-space estimate, inverse Fourier-transformed and coil channels reduced via the estimated sensitivity maps. The form of the reduction operator can be seen in line 255. In particular, the input has shape $2s \times h \times w$, that is the $c$ coil channels have been combined for each slice.
> >
> > Detailed comments 9.: The factor of 10 comes from the approximately factor 3 difference along each spatial dimension, resulting in an approximately 10-folds difference in the number of pixels.

---

### Author Response · Authors · 2022-08-02
**Summary of rebuttal**

We would like to thank all reviewers for their effort going over our paper in such depth and giving us very insightful feedback. We are glad that the paper has overall good reception. Reviewers generally think that “the paper is very well-written” and mention that our “novel approach paves way for applied MRI models with hybrid convolutional-transformer architectures” and that “having a new state-of-the-art for accelerated MRI reconstruction is also highly appreciated”. We did our best to address all issues raised by the reviewers and updated the paper reflecting our response to the comments. Modified/new portions of the paper are highlighted in blue font.
An overview of the changes:
- We moved the appendix from the supplementary to the back of the main paper.
- We added an extra ablation study on the effect of Adjacent Slice Reconstruction on VarNet performance (Appendix D).
- We added an extra ablation study on HUMUS-Net’s performance scaling with the number of cascades (Appendix C).
- We added an extra discussion on the intuition of iterative denoising with plots of intermediate reconstructions in HUMUS-Net (Appendix E).
- We added many extra details such as results in more metrics (Appendix G), an overview of Vision Transformer terminology (Appendix F) and a more detailed plot of patch operations (Appendix H)
- We did our best to rephrase and clarify the main text following reviewers’ comments.

We believe that these changes further strengthened our paper. That said, we are open for further discussion and feedback.

---

> ### Author Response · Authors · 2022-08-09
> **Final comment and some changes**
>
> We appreciate the reviewers’ feedback during the discussion period. We did our best to answer all comments within the short time frame. Additionally, following feedback from reviewer **pRk7** and **4jKX** we extended the Appendix with experiments on different acceleration factors ranging from 4x to 12x that can be found in Appendix I. We hope that we were able to convince the reviewers about the merit and significance of HUMUS-Net.

---

### Meta-Review · Area_Chair_v7Ax · 2022-08-25

**Recommendation:** Accept
**Confidence:** Certain

**Metareview:**

Four reviewers generally favour accepting the paper, and I agree. The authors have done a good job of addressing the most pressing concerns of the reviewers in the rebuttal period.

**Award:**

No

---

### Decision · Program_Chairs · 2022-09-14

Accept